Evaluation of total immunoglobulin G and subclass antibodies in an enzyme-linked immunosorbent assay for serodiagnosis of human amebic liver abscess

Janwan Penchom 1 2
Sadaow Lakkhana 3 4
Rodpai Rutchanee 3 4
Yamasaki Hiroshi 5 6
Luvira Vor 7
Sukeepaisarnjaroen Wattana 8
Kitkhuandee Amnat 7
Paonariang Krisada 7
Sanpool Oranuch 3 4
Boonroumkaew Patcharaporn 3 4
Thanchomnang Tongjit 4 9
Mita Toshihiro 6
Intapan Pewpan M. 3 4
Maleewong Wanchai wanch_ma@kku.ac.th 3 4
1 Department of Medical Technology, School of Allied Health Sciences, Walailak University , Nakhon Si Thammarat , Thailand
2 Hematology and Transfusion Science Research Center, Walailak University , Nakhon Si Thammarat , Thailand
3 Department of Parasitology, Faculty of Medicine, Khon Kaen University , Khon Kaen , Thailand
4 Mekong Health Science Research Institute, Khon Kaen University , Khon Kaen , Thailand
5 Department of Parasitology, National Institute of Infectious Diseases , Tokyo , Japan
6 Department of Tropical Medicine and Parasitology, Juntendo University School of Medicine , Tokyo , Japan
7 Department of Surgery, Faculty of Medicine, Khon Kaen University , Khon Kaen , Thailand
8 Department of Medicine, Faculty of Medicine, Khon Kaen University , Khon Kaen , Thailand
9 Faculty of Medicine, Mahasarakham University , Maha Sarakham , Thailand
Braga Erika
Electronic publication date: 2022 Sep 29
Publication date: 2022
Volume: 10
Electronic Location ID: e14085
Received 2022 Jun 3; Accepted 2022 Aug 29
Copyright: ©2022 Janwan et al.
Copyright year: 2022
Copyright holder: Janwan et al.
License: This is an open access article distributed under the terms of the Creative Commons Attribution License, which permits unrestricted use, distribution, reproduction and adaptation in any medium and for any purpose provided that it is properly attributed. For attribution, the original author(s), title, publication source (PeerJ) and either DOI or URL of the article must be cited.
License URL: https://creativecommons.org/licenses/by/4.0/

Keywords: Amebic liver abscess, Entamoeba histolytica, Enzyme-linked immunosorbent assay, Immunoglobulin G subclasses, Serodiagnosis

Funding: National Science, Research and Innovation Fund, Thailand (WM) Khon Kaen University, Research and Graduate Studies Affairs Faculty of Medicine, Khon Kaen University RG63301 This study was supported by the National Science, Research and Innovation Fund, Thailand (WM); Khon Kaen University, Research and Graduate Studies Affairs, (PJ, OS, PMI and WM, Research Program); and the Faculty of Medicine, Khon Kaen University (PMI and WM, grant number RG63301). The funders had no role in study design, data collection and analysis, decision to publish, or preparation of the manuscript.

==============================
Background

Amebic liver abscess (ALA) caused by Entamoeba histolytica is usually diagnosed based on its clinical symptoms, medical imaging abnormalities of the liver, and serological tests, the most common being the enzyme-linked immunosorbent assay (ELISA). For more than three decades, no investigation has evaluated the diagnostic performance of immunoglobulin G (IgG) subclasses in the serodiagnosis of ALA. Herein, we assessed the efficiencies of anti-amebic IgG and IgG subclasses for diagnosing ALA.

Methods

A serological ELISA-based test was performed to assess its diagnostic performance using a total of 330 serum samples from ALA patients (n = 14), healthy individuals (n = 40), and patients with other diseases (n = 276).

Results

ELISA targeting the total IgG antibody to E. histolytica antigen exhibited 100% sensitivity 95% CI [76.8–100.0] and 97.8% specificity 95% CI [95.5–99.1], whereas the assay targeting IgG1 showed the same sensitivity (100% 95% CI [76.8–100.0]) and a slightly higher specificity (99.1% 95% CI [97.3–99.8]). The other IgG subclasses (IgG2, IgG3, and IgG4) displayed a lower sensitivity and specificity. The sensitivity and specificity did not significantly differ between tests measuring total IgG and IgG1 (Exact McNemar’s test; p > 0.05), with a concordance of 98.2%, represented by a Cohen’s kappa of 0.83 (p < 0.001), indicating almost perfect agreement.

Conclusion

ELISA targeting IgG1 can provide valuable information to clinicians in differentiating ALA from other parasitic diseases, cancers, cirrhosis, and viral hepatitis. However, enzyme-conjugated anti-human total IgG is cheaper than anti-human IgG subclasses. Therefore, we suggest that total IgG-based ELISA is sufficient for the routine serodiagnosis of human ALA and possibly other clinical manifestations of invasive amebiasis.

Introduction

Amebiasis, an infection caused by Entamoeba histolytica, is the second leading cause of death from protozoan infections worldwide (Stanley Jr, 2003; Lozano et al., 2012), affecting an estimated 50 million people and causing the death of 40,000–100,000 people yearly (Shirley et al., 2018). The infection is frequently transmitted to humans via the fecal-oral route, which is prevalent in areas with poor sanitation, or through oral-anal sexual contact (Tanyuksel & Petri Jr, 2003; Shirley et al., 2018). Most infected people are asymptomatic, but some develop invasive amebiasis, which is becoming more common among homosexual men and acquired immunodeficiency syndrome patients (Takeuchi et al., 1989; Huang et al., 2020).

The most common extraintestinal manifestation is an amebic liver abscess (ALA), which is associated with significant morbidity and mortality (Alavi, 2007; Kannathasan et al., 2018; Marenga et al., 2019). Trophozoites of E. histolytica travel from the intestine to the liver via the portal vein and produce hepatic inflammation followed by necrosis, leading to an abscess. ALA can be diagnosed by detecting motile trophozoites in liver pus; however, this requires a relatively invasive procedure and is not a sensitive diagnostic method. Polymerase chain reaction (PCR) assay, the gold standard method, helps diagnose ALA if aspirated pus is available, but this technique is costly and only used in some laboratories, particularly in developed countries, where infrastructure and skillful staff are available (Zaman et al., 2000; Tanyuksel & Petri Jr, 2003; Ryan, Paparini & Oskam, 2017; Saidin, Othman & Noordin, 2019). Therefore, in practice, the diagnosis of ALA is based on clinical features, history of living in or traveling to endemic regions, imaging techniques such as ultrasound (US), computerized tomography (CT), and magnetic resonance imaging (MRI), as well as serological tests (Wong et al., 2017), like immunofluorescence (IF) tests (Garcia et al., 1982; Jackson, Anderson & Simjee, 1984), indirect hemagglutination assays (IHA) (Knobloch & Mannweiler, 1983; Hung et al., 1999; Dhanalakshmi, Meenachi & Parija, 2016), enzyme immunoassays (Yang & Kennedy, 1979; Hock et al., 1989; Shetty et al., 1990; Knappik, Borner & Jelinek , 2005; Wong et al., 2017; Beyls et al., 2018), and lateral flow immunoassays (Saidin et al., 2014; Tachibana et al., 2018; Noordin et al., 2020).

The enzyme-linked immunosorbent assay (ELISA) is still the most common test in many countries for diagnosing ALA due to its high sensitivity, specificity, ease of use, and cost-effectiveness. In addition, detecting antibodies can be beneficial when fecal microscopy or antigen detection tests are ineffective for diagnosis. In endemic settings, however, significant background measurements are produced; consequently, the original protocol’s cut-off may need to be adjusted for each individual situation (Fotedar et al., 2007; Saidin, Othman & Noordin, 2019), particularly by including healthy volunteer samples in each area for baseline subtraction and evaluating a larger number of other infectious disease samples as preliminary data for describing awareness prior to clinical use. To discriminate between previous and present infections, the best specimens, like pus or serum, should be submitted to a molecular reference laboratory for identifying the infectious agent (Zaman et al., 2000; Ghelfenstein-Ferreira et al., 2020). Nowadays, in developing countries with limited resources, where PCR is not routinely conducted in the clinical laboratory, positive antibody testing along with a patient’s clinical manifestations, risk history, other laboratory findings, and/or abdominal imaging are useful for diagnosis (Fotedar et al., 2007; Saidin, Othman & Noordin, 2019).

For ELISA, most of the circulating E. histolytica-specific antibody detection tests indicate using immunoglobulin G (IgG) in the detection system (Yang & Kennedy, 1979; Knappik, Borner & Jelinek, 2005; Fotedar et al., 2007; Wong et al., 2017; Beyls et al., 2018; Saidin, Othman & Noordin, 2019; Watanabe et al., 2021) because invasive amebic infections provoke a strong humoral response, especially involving the IgG class (Hock et al., 1989; Shetty et al., 1990). Anti-amebic IgG and its subclasses were determined by ELISA in the sera of ALA patients and compared with those from normal healthy controls. The positive rates of anti-amebic antibodies for total IgG, IgG1, IgG2, IgG3, and IgG4 were 97.7%, 79.6%, 31.8%, 31.8%, and 68.2%, respectively (Hock et al., 1989). However, to ensure the system’s diagnostic effectiveness, the ELISA should include sera from patients with other diseases.

For more than three decades, no investigation has evaluated the diagnostic performance of IgG subclasses in the serodiagnosis of ALA. Thus, this study aimed to evaluate the diagnostic efficiencies of anti-amebic IgG and IgG subclasses, determined by ELISA, in sera of patients with ALA, with other parasitic diseases, and other conditions such as liver, bile duct, brain, and colon cancers, cirrhosis, and viral hepatitis.

Materials & Methods

Human sera

All human sera used in this study, except samples from ALA and clonorchiasis patients, were obtained from the serum bank of the Department of Parasitology, Faculty of Medicine, Khon Kaen University, Khon Kaen, Thailand. Sera from amebiasis and clonorchiasis patients were provided by the Department of Tropical Medicine and Parasitology, Juntendo University School of Medicine, Tokyo, Japan and the Department of Parasitology, National Institute of Infectious Diseases, Tokyo. These sera were categorized into three groups: (1) negative control samples from healthy volunteers from northeastern Thailand (n = 40), which were proven to be free of any intestinal parasitic infections at the time of blood collection; (2) samples from cases of ALA caused by E. histolytica (n = 14), which were described in detail together with the clinical courses and diagnostic evidence of ALA (Table S1); and (3) samples from patients diagnosed with other diseases (n = 276). No mixed parasitic infection sera were used in this study. The frozen sera stocks of fourteen Japanese ALA patients whose illnesses that occurred between 1980 and 1997 (Table S1) were used. More than two decades ago, PCR assays were not well-developed and were not used to confirm ALA diagnoses in those cases. The pus aspirate samples, a specimen of choice for PCR assay of all ALA cases, were no longer available to us. Therefore, the diagnosis of ALA in this study was based on the medical records of clinical presentations, such as fever and right hypochondrial pain, amebic liver abscess, US and CT imaging findings, and serological confirmation tests against E. histolytica antigen, like ELISA, IF test, IHA, Ouchterlony’s immunodiffusion test, and complement fixation test (Table S1) which still being used these guidelines today (Tachibana et al., 2018; Watanabe et al., 2021). Serological assays for diagnosing ALA be useful in industrialized countries where E. histolytica infections are uncommon (Fotedar et al., 2007), including Japan. The non-infected status of healthy volunteers, and the infected status of individuals with ascariasis, hookworm infection, trichuriasis, capillariasis, opisthorchiasis, fascioliasis, and paragonimiasis were confirmed by parasitological examination of stool samples using the formalin ethyl acetate concentration technique (Elkins, Haswell-Elkins & Anderson, 1986). Strongyloidiasis was confirmed by the agar-plate culture method (Koga et al., 1991). Trichinellosis was confirmed by detection of intramuscular larvae and serological methods (Morakote et al., 1991). Angiostrongyliasis was diagnosed by serological methods with clinical manifestations (Somboonpatarakun et al., 2020). Gnathostomiasis was confirmed by serological methods with clinical manifestations and history of dietary preferences (Janwan et al., 2016). Clonorchiasis was confirmed using the modified Kato-Katz technique (Katz, Chaves & Pellegrino, 1972) and ELISA (Yu et al., 2003). Cysticercosis was confirmed by CT and serological methods (Intapan et al., 2008). Schistosomiasis was confirmed by the modified Kato-Katz technique (Katz, Chaves & Pellegrino, 1972). Sparganosis was diagnosed using histopathological investigation and PCR (Boonyasiri et al., 2014). Blastocystosis and giardiasis were confirmed by the direct simple smear technique (World Health Organization, 1997). Malaria and filariasis were confirmed by the Giemsa-stained blood smear technique (World Health Organization, 1997). Viral hepatitis was diagnosed using the electrochemiluminescence immunoassay in the cobas 8000 modular analyzer (Roche Diagnostics, Tokyo, Japan). Cirrhosis, hepatocellular carcinoma, cholangiocarcinoma, brain cancer, and colon cancer were confirmed by blood-based tumor biomarkers, dynamic contrast-enhanced CT, MRI, and histopathological analysis according to the specialist surgeon’s diagnosis at Srinagarind Hospital, Faculty of Medicine, Khon Kaen University. Pooled positive and negative reference sera were prepared by mixing equal volumes of sera from 10 ALA patients and 10 healthy volunteers, respectively. These pooled reference sera were further used as control sera for within-day and between-day precisions in ELISA plates.

The study protocol was approved by the Human Research Ethics Committee of Khon Kaen University (HE641225, approved 23 April 2021) and the study was conducted in accordance with the Declaration of Helsinki and the ICH Good Clinical Practice Guidelines.

Parasite antigen preparation

The E. histolytica HK-9 strain was cultivated in Diamond’s medium as per previously published protocols (Diamond, Harlow & Cunnick, 1978). Briefly, the trophozoites, after 48–72 h culture, were chilled to dislodge the parasites and washed thrice with normal saline solution (NSS) by centrifuging at 120 g at room temperature (RT) for 5 min. The cell sediment was resuspended in an appropriate volume of NSS to give approximately 107 cells/mL. The preparation was then sonicated with an ultrasonic disintegrator (Sonics & Materials, Newtown, CT) and centrifuged at 10,000 g for 30 min at 4 °C. The supernatant was dialyzed against distilled water containing proteinase inhibitors (Roche Applied Science, Basel, Switzerland) and stored at −20 °C until further used as crude E. histolytica HK-9 antigen in the ELISA. The protein content of the antigen was determined using the standard method (Lowry et al., 1951).

ELISA

A standard ELISA protocol was followed. Antigen, serum, conjugate, and substrate were used at volumes of 100 µL/well and the optimum condition to be finally used in ELISA was determined by checkerboard titration. A 96-well microtiter plate (Corning Incorporated, Kennebunk, ME) was coated overnight with the antigen at an optimum concentration of 2.5 µg/mL (for total IgG) and 5.0 µg/mL (for IgG subclasses) in carbonate buffer, pH 9.6, at 4 °C. The plates were washed five times with 0.05% Tween 20 in 0.1 M phosphate-buffered saline, pH 7.4 (PBST). After blocking nonspecific binding sites with 300 µL of PBST-3% bovine serum albumin (BSA) for 1 h at RT, the plate was then washed with PBST, incubated with diluted serum in PBST-1% BSA for 1 h at 37 °C, and washed again with PBST. The optimum serum dilution was 1:800 for detecting total IgG and 1:100 for IgG subclasses. This was followed by incubating with either horseradish peroxidase (HRP)-conjugated monoclonal anti-human IgG (Invitrogen Corporation, Camarillo, CA) diluted 1:100,000 for total IgG, or HRP-conjugated anti-human IgG subclass antibodies diluted 1:6,000 for IgG1 (Zymed, South San Francisco, CA), 1:4,000 for IgG2 (Zymed), 1:500 for IgG3 (Invitrogen), and 1:1,000 for IgG4 (Invitrogen) for 1 h at 37 °C. After thorough washing, o-phenylenediamine dihydrochloride (Sigma-Aldrich, St. Louis, MO) in citrate phosphate buffer, pH 5.0, was used as a substrate and incubated for 30 min at RT in the dark. The enzyme reaction was stopped by adding 50 µL of 8 N sulfuric acid. The optical density (OD) was measured at 492 nm using a microplate reader (Tecan, Salzburg, Austria). Duplicates of each serum sample were included in the same plate to ensure reproducibility of the tests. The precision of the ELISA was investigated by performing the test on different days using the same pooled positive and negative sera, and the same batch of antigen under the same conditions. Consistent data were obtained from all the tests indicating no day-to-day variation.

Data analysis

Data obtained from the analyses of clinical samples were recorded in Microsoft Excel. All statistical analyses were conducted using the statistical package Stata version 10.1 (StataCorp LLC, College Station, TX). Continuous variables were compared using the nonparametric Kruskal–Wallis method. Quantitative data were presented as mean ± standard deviation. The cut-off values that gave the best balance of sensitivity and specificity for ELISA were established using the receiver operating characteristic (ROC) curve with sera from ALA patients and from healthy individuals. The standard diagnostic indices, including accuracy, sensitivity, specificity, and likelihood ratios, were calculated as previously described Galen (1980). The variables measured were the number of true positives (TP), the number of true negatives (TN), the number of false positives (FP), and the number of false negatives (FN). The test accuracy, defined as the proportion of all tests that gave a correct result, (TP + TN)/number of all tests. The sensitivity was calculated as TP/(TP + FN) and the specificity as TN/(TN + FP). The positive likelihood ratio was calculated as sensitivity/(100 − specificity) and the negative likelihood ratio as (100 − sensitivity)/specificity. The sensitivity, specificity, and cross-reactivity of total IgG were compared with those of the IgG subclasses using McNemar’s test. The total concordance was calculated using Cohen’s kappa test. Interpretations of kappa values (κ) were graded as almost perfect agreement (κ = 0.81–1.00), substantial (κ = 0.61–0.80), moderate (κ = 0.41–0.60), fair (κ = 0.21–0.40), slight (κ = 0.00–0.20), and poor agreement (κ <0.00) (Landis & Koch, 1977).

Results

We demonstrated the responses of total IgG and IgG subclasses against the E. histolytica HK-9 antigen (Table 1, Fig. 1). The mean OD values for total IgG and IgG subclasses in the ALA group were significantly higher than those for diseases other than ALA and for the healthy controls (p < 0.001). According to the ROC analysis, the best cut-off OD values were 0.412, 0.079, 0.200, 0.093, and 0.139 for total IgG, IgG1, IgG2, IgG3, and IgG4, respectively. The sensitivities for detecting total IgG, IgG1, IgG2, IgG3, and IgG4 were 100%, 100%, 92.9%, 71.4%, and 85.7%, respectively (Table 2). Apart from IgG1 (99.1% specificity), the individual IgG subclasses had a lower specificity than that of the total IgG antibody (97.8% specificity) (Table 2). As presented in Table 1, some cross-reactions occurred. As for the total IgG response, cross-reactions were observed in one case each of hookworm infection, capillariasis, clonorchiasis, schistosomiasis, and blastocystosis, as well as two cases of fascioliasis. For the IgG1 response, cross-reactions were observed in one case each of hookworm infection, clonorchiasis, and blastocystosis. No cross-reactions for both the above-mentioned IgG responses were observed in sera from patients with viral hepatitis, cirrhosis, and cancers. Overall, we demonstrated that IgG1 had the greatest diagnostic potential. The sensitivity and specificity of tests measuring total IgG and IgG1 did not differ significantly (Exact McNemar’s test; p > 0.05), with a concordance of 98.2% represented by a Cohen’s kappa of 0.83 (p <  0.001), indicating almost perfect agreement (Table 2).

Table 1 Optical density (OD) values of total IgG and IgG subclass antibodies to E. histolytica HK-9 antigen-based ELISA and percentage above the cut-off value.

Type of serum (no. examined)	Mean OD ± SD (% Positivity)	
	Total IgG	IgG1	IgG2	IgG3	IgG4	
Healthy controls (40)	0.139 ± 0.060 (0)	0.027 ± 0.011 (0)	0.060 ± 0.021 (0)	0.031 ± 0.024 (0)	0.051 ± 0.036 (0)	
Amebic liver abscess (14)	1.170 ± 0.413 (100.0)	0.869 ± 0.622 (100.0)	0.465 ± 0.272 (92.9)	0.279 ± 0.251 (71.4)	0.614 ± 0.733 (85.7)	
Ascariasis (10)	0.112 ± 0.029 (0)	0.027 ± 0.006 (0)	0.028 ± 0.017 (0)	0.012 ± 0.007 (0)	0.028 ± 0.029 (0)	
Hookworm infection (15)	0.183 ± 0.092 (6.7)	0.036 ± 0.044 (6.7)	0.072 ± 0.067 (6.7)	0.044 ± 0.041 (13.3)	0.068 ± 0.057 (13.3)	
Trichuriasis (15)	0.130 ± 0.066 (0)	0.032 ± 0.017 (0)	0.070 ± 0.079 (6.7)	0.044 ± 0.054 (13.3)	0.049 ± 0.049 (6.7)	
Capillariasis (10)	0.197 ± 0.122 (10.0)	0.019 ± 0.015 (0)	0.078 ± 0.060 (10.0)	0.069 ± 0.109 (10.0)	0.226 ± 0.132 (70.0)	
Strongyloidiasis (15)	0.140 ± 0.065 (0)	0.036 ± 0.015 (0)	0.055 ± 0.060 (6.7)	0.028 ± 0.027 (6.7)	0.166 ± 0.524 (6.7)	
Trichinellosis (15)	0.149 ± 0.048 (0)	0.020 ± 0.009 (0)	0.080 ± 0.042 (0)	0.055 ± 0.052 (13.3)	0.069 ± 0.049 (6.7)	
Angiostrongyliasis (10)	0.159 ± 0.030 (0)	0.027 ± 0.009 (0)	0.053 ± 0.024 (0)	0.081 ± 0.132 (30.0)	0.033 ± 0.025 (0)	
Gnathostomiasis (15)	0.146 ± 0.063 (0)	0.017 ± 0.009 (0)	0.058 ± 0.021 (0)	0.090 ± 0.091 (33.3)	0.079 ± 0.131 (13.3)	
Clonorchiasis (20)	0.221 ± 0.093 (5.0)	0.036 ± 0.020 (5.0)	0.083 ± 0.034 (0)	0.055 ± 0.034 (10.0)	0.173 ± 0.182 (45.0)	
Opisthorchiasis (20)	0.157 ± 0.061 (0)	0.025 ± 0.013 (0)	0.074 ± 0.062 (5.0)	0.069 ± 0.080 (20.0)	0.061 ± 0.076 (15.0)	
Fascioliasis (10)	0.263 ± 0.156 (20.0)	0.028 ± 0.013 (0)	0.094 ± 0.047 (0)	0.090 ± 0.080 (30.0)	0.547 ± 0.483 (90.0)	
Paragonimiasis (10)	0.191 ± 0.065 (0)	0.030 ± 0.014 (0)	0.142 ± 0.091 (10.0)	0.060 ± 0.042 (20.0)	0.322 ± 0.416 (60.0)	
Cysticercosis (15)	0.182 ± 0.109 (0)	0.027 ± 0.009 (0)	0.140 ± 0.193 (13.3)	0.145 ± 0.237 (26.7)	0.118 ± 0.129 (26.7)	
Schistosomiasis (14)	0.211 ± 0.102 (7.1)	0.028 ± 0.013 (0)	0.051 ± 0.024 (0)	0.270 ± 0.372 (50.0)	0.126 ± 0.152 (28.6)	
Sparganosis (6)	0.119 ± 0.038 (0)	0.016 ± 0.006 (0)	0.117 ± 0.074 (16.7)	0.019 ± 0.016 (0)	0.204 ± 0.268 (33.3)	
Blastocystosis (15)	0.183 ± 0.162 (6.7)	0.064 ± 0.144 (6.7)	0.091 ± 0.059 (6.7)	0.067 ± 0.091 (26.7)	0.128 ± 0.213 (13.3)	
Giardiasis (15)	0.150 ± 0.061 (0)	0.020 ± 0.012 (0)	0.053 ± 0.023 (0)	0.183 ± 0.444 (26.7)	0.017 ± 0.014 (0)	
Viral hepatitis (4)	0.243 ± 0.083 (0)	0.044 ± 0.021 (0)	0.129 ± 0.046 (0)	0.077 ± 0.033 (25.0)	0.175 ± 0.146 (50.0)	
Cirrhosis (4)	0.151 ± 0.062 (0)	0.036 ± 0.025 (0)	0.077 ± 0.047 (0)	0.052 ± 0.032 (25.0)	0.137 ± 0.095 (25.0)	
Hepatocellular carcinoma (4)	0.266 ± 0.080 (0)	0.036 ± 0.014 (0)	0.132 ± 0.029 (0)	0.146 ± 0.122 (50.0)	0.186 ± 0.089 (75.0)	
Cholangiocarcinoma (4)	0.116 ± 0.042 (0)	0.012 ± 0.004 (0)	0.096 ± 0.025 (0)	0.034 ± 0.035 (0)	0.051 ± 0.022 (0)	
Brain cancer (4)	0.110 ± 0.039 (0)	0.009 ± 0.007 (0)	0.054 ± 0.006 (0)	0.012 ± 0.006 (0)	0.054 ± 0.048 (0)	
Colon cancer (4)	0.165 ± 0.066 (0)	0.022 ± 0.013 (0)	0.078 ± 0.017 (0)	0.196 ± 0.302 (25.0)	0.040 ± 0.018 (0)	
Malaria (Plasmodium falciparum infection) (10)	0.209 ± 0.080 (0)	0.023 ± 0.016 (0)	0.057 ± 0.024 (0)	0.070 ± 0.033 (20.0)	0.049 ± 0.042 (10.0)	
Malaria (Plasmodium vivax infection) (10)	0.120 ± 0.055 (0)	0.008 ± 0.003 (0)	0.043 ± 0.017 (0)	0.027 ± 0.024 (0)	0.015 ± 0.014 (0)	
Filariasis (2)	0.118 ± 0.029 (0)	0.003 ± 0.000 (0)	0.037 ± 0.002 (0)	0.059 ± 0.000 (0)	0.010 ± 0.000 (0)	

Figure 1 Absorbance scattergrams with 95% confidence intervals of total IgG and IgG subclasses according to different serum groups.

Group I, healthy control group (n = 40); Group II, confirmed amebic liver abscess (ALA) cases (n = 14); and Group III, diseases other than ALA (n = 276).

Table 2 Diagnostic values for the detection of amebic liver abscess and test for agreement between total IgG and various IgG subclass antibodies by ELISA.

	Diagnostic values (95% CI)	
	Total IgG	IgG1	IgG2	IgG3	IgG4	
Accuracy (%)	97.9 (95.7–99.1)	99.1 (97.4–99.8)	96.7 (94.1–98.3)	82.7 (78.2–86.6)	81.2 (76.6–85.3)	
Sensitivity (%)	100.0 (76.8–100.0)	100.0 (76.8–100.0)	92.9 (66.1–99.8)	71.4 (41.9–91.6)	85.7 (57.2–98.2)	
Specificity (%)	97.8 (95.5–99.1)	99.1 (97.3–99.8)	96.8 (94.3–98.5)	83.2 (78.6–87.2)	81.0 (76.2–85.2)	
ROC area	0.989 (0.981–0.997)	0.995 (0.990–1.000)	0.948 (0.878–1.000)	0.773 (0.649–0.898)	0.834 (0.736–0.931)	
Positive likelihood ratio	45.1 (21.7–93.9)	105.0 (34.2–325.0)	29.3 (15.7–54.9)	4.3 (2.8–6.4)	4.5 (3.3–6.2)	
Negative likelihood ratio	0.0	0.0	0.1 (0.0–0.5)	0.3 (0.2–0.8)	0.2 (0.0–0.6)	
Test for agreement between total IgG and the IgG subclasses	
Exact McNemar	–	0.22	0.79	0.00	0.00	
Observed agreement	–	98.2	95.8	83.6	82.1	
Cohen’s kappa	–	0.83	0.66	0.29	0.30	
p-value	–	0.00	0.00	0.00	0.00	
Notes.

CI confidence interval

ROC receiver operating characteristic

Discussion

Human ALA is diagnosed based on clinical presentation with relevant epidemiology, abnormalities seen in medical images of the liver, and serological findings (Tanyuksel & Petri Jr, 2003). ELISA is the most commonly used serological screening test in the world (Fotedar et al., 2007; Saidin, Othman & Noordin, 2019), superior due to its high-throughput potential, low cost, high reliability, ease of standardization, and an acceptable balance between sensitivity and specificity. To our knowledge, for more than three decades, no studies have evaluated the efficacy of total IgG and IgG subclasses for diagnosing ALA using sera from patients with various diseases. Cross-reactivity should be examined to understand other possibilities leading to a positive result. These findings are crucial for clinicians working in areas where operating rooms, advanced radiological equipment, and laboratories are not fully established yet, necessitating the use of serodiagnosis to support the main diagnosis.

The present study was designed to assess the diagnostic potential of ELISA, targeting total IgG and IgG subclasses produced against crude soluble antigens of E. histolytica. Our IgG-based ELISA showed a high diagnostic sensitivity (100%) and specificity (97.8%), comparable to the values of 80–100% sensitivities and 93–100% specificities reported for commercial and in-house IgG ELISA kits in cases of ALA (Tanyuksel & Petri Jr, 2003; Knappik, Borner & Jelinek , 2005; Wong et al., 2017; Beyls et al., 2018; Saidin, Othman & Noordin, 2019). Interestingly, group 3 contains sera from patients with significant parasitic diseases prevalent in the developing world, implying that these patients are hugely affected by protozoa, including E. histolytica, on a regular basis. The positive rate for E. histolytica detected by total IgG ELISA was only 2.5% (7 out of 276 sera) in group 3, which is not very high compared with the average for developing nations. These results demonstrate that our IgG-based ELISA method is valuable in the context of a real-world investigation conducted in a developing nation. Our in-house ELISA has two significant flexibility advantages over commercially available assays: (1) shelf life, freshly prepared ELISA components are dependent on the physician’s estimated monthly requests, and (2) the assay conditions, including serum dilution, conjugate dilution, type of conjugate and blocking buffer, etc., can be optimized to each endemic zone.

Cross-reactions in the IgG ELISA test were observed in some samples from patients infected with other parasites: hookworm infection, capillariasis, clonorchiasis, fascioliasis, schistosomiasis, and blastocystosis. The occasional cross-reactions with these other parasite species do not pose a significant problem in the clinical setting. Hookworm infection, capillariasis, and blastocystosis normally show a diagnostic stage in fecal specimens and have clinical symptoms that differ from those of ALA. Although clonorchiasis, fascioliasis, and schistosomiasis can elicit clinical symptoms and liver inflammation/abscess similar to ALA, they also present with a history of infection, microscopic examination of the stool, peripheral blood eosinophilia, medical imaging abnormalities of the liver, and medication responses that are different from those in ALA (Stanley Jr, 2003; Kaya, Beştaş & Cetin, 2011; Cavalcanti, de Araujo-Neto & Peralta, 2015; Gowda, 2015; Peters, Burkert & Grüner, 2021). Moreover, cross-reactions (20.0%) with fascioliasis were found in ELISA targeting total IgG. If total IgG yields a positive result in regions where ALA and fascioliasis co-exist, we suggest evaluating peripheral blood eosinophilia—if it is abnormally high, fascioliasis is suspected; if not, ALA is suspected (Stanley Jr, 2003; Fotedar et al., 2007; Shirley et al., 2018; Peters, Burkert & Grüner, 2021).

During infection, different IgG subclasses are produced but, as we have shown here, all of them were not equally elevated in the humoral response to ALA. This situation is likely to be consistent and predictable (Hock et al., 1989; Kaur et al., 2004). In this study, IgG1 appeared to be the most suitable IgG subclass for ALA diagnosis, which corroborates previous findings (Hock et al., 1989; Kaur et al., 2004). Thus, comparing the Cohen’s kappa test with antibody test kits routinely used with the IgG1-based ELISA assay might be interesting. Moreover, our results suggest that IgG3 and IgG4 subclasses are poor diagnostic markers for ALA because of high cross-reactivity with other diseases. In addition, serial studies to observe the levels of amebic IgG and IgG subclasses in ALA patients, from the onset of symptoms to successful cure, should yield valuable kinetic data regarding the levels of these IgGs during the course of infection.

We realize that ELISA is time consuming. However, as a routine laboratory service, this is the first step taken in developing countries, like Thailand and nearby Asian countries. Our in-house ELISA can be manipulated more easily than the PCR assay. Cost also significantly limits the use of current PCR methods, particularly in developing nations. In such nations, positive antibody testing, along with a patient’s clinical manifestations, risk history, other laboratory findings, and/or abdominal imaging, are currently used for diagnosis (Fotedar et al., 2007; Saidin, Othman & Noordin, 2019). Therefore, in this study, we put enormous effort toward presenting empirical evidence for the efficacy of well-optimized ELISA-based anti-amebic antibody detection, particularly by including healthy volunteer samples in the study area for baseline subtraction and evaluating many other infectious disease samples. This serves as preliminary data for describing awareness before clinical use. We hope these findings can be applied in developing countries. As a next step, we intend to develop an in-house lateral flow immunoassay-based approach: a user-friendly and rapid diagnostic platform necessary for point-of-care diagnosis in resource-limited settings.

The present study has some limitations. First, our study was carried out using a small sample size of ALA patients, and the PCR assay was not used to confirm the ALA diagnosis. So, there was some bias in estimating the sensitivity for ALA patients. Second, the sera were obtained from banks in Thailand and Japan; the results may not apply to other populations. Third, cross-reactions should be investigated with pyogenic liver abscess, leishmaniasis, and other infectious diseases primarily found in tropical regions. Addressing these limitations will strengthen the clinical laboratory application of this study.

Conclusions

No previous studies have investigated the diagnostic effectiveness of total IgG and IgG subclasses in ALA. We found that IgG1 is an excellent diagnostic antibody for ALA. ELISA targeting this subclass of IgG can provide valuable information to clinicians by differentiating ALA from other parasitic diseases, cancers, cirrhosis, and viral hepatitis. However, enzyme-conjugated anti-human IgG subclasses are quite costly. Considering no significant differences in sensitivity and specificity between IgG1 and total IgG detection and almost perfect agreement between them, we recommend that a total IgG-based ELISA is sufficient for the routine serodiagnosis of human ALA and possibly other clinical manifestations of invasive amebiasis.

We would like to thank David Blair for editing the English presentation of this paper via the Publication Clinic Khon Kaen University, Thailand. Special thanks to three referees for valuable comments and suggestions to improve the manuscript. The authors also would like to thank Enago for the English language review.

Additional Information and Declarations

Competing Interests

Author Contributions

Human Ethics

Data Availability

Supplemental Information

Supplemental Information 1 The raw optical density (OD) values of total IgG and IgG subclasses in all serum samples were analyzed in this study

Click here for additional data file.

Supplemental Information 2 Demographic and clinical information of Japanese amebic liver abscess (ALA) patients examined in this study

Click here for additional data file.

The authors declare there are no competing interests.

Penchom Janwan conceived and designed the experiments, performed the experiments, analyzed the data, prepared figures and/or tables, authored or reviewed drafts of the article, and approved the final draft.

Lakkhana Sadaow performed the experiments, analyzed the data, prepared figures and/or tables, authored or reviewed drafts of the article, and approved the final draft.

Rutchanee Rodpai performed the experiments, analyzed the data, authored or reviewed drafts of the article, and approved the final draft.

Hiroshi Yamasaki conceived and designed the experiments, authored or reviewed drafts of the article, and approved the final draft.

Vor Luvira performed the experiments, authored or reviewed drafts of the article, contributed materials, and approved the final draft.

Wattana Sukeepaisarnjaroen performed the experiments, authored or reviewed drafts of the article, contributed materials, and approved the final draft.

Amnat Kitkhuandee performed the experiments, authored or reviewed drafts of the article, contributed materials, and approved the final draft.

Krisada Paonariang performed the experiments, authored or reviewed drafts of the article, contributed materials, and approved the final draft.

Oranuch Sanpool performed the experiments, authored or reviewed drafts of the article, and approved the final draft.

Patcharaporn Boonroumkaew performed the experiments, analyzed the data, prepared figures and/or tables, authored or reviewed drafts of the article, and approved the final draft.

Tongjit Thanchomnang performed the experiments, authored or reviewed drafts of the article, and approved the final draft.

Toshihiro Mita performed the experiments, analyzed the data, prepared figures and/or tables, authored or reviewed drafts of the article, contributed reagents and materials, and approved the final draft.

Pewpan M. Intapan conceived and designed the experiments, authored or reviewed drafts of the article, contributed reagents and materials, and approved the final draft.

Wanchai Maleewong conceived and designed the experiments, authored or reviewed drafts of the article, contributed reagents and materials, and approved the final draft.

The following information was supplied relating to ethical approvals (i.e., approving body and any reference numbers):

The study protocol was approved by the Human Research Ethics Committee of Khon Kaen University (HE641225, approved 23 April 2021) in accordance with the Declaration of Helsinki and the ICH Good Clinical Practice Guidelines. Informed consent was obtained from all adult participants, parents or legal guardians of minors.

The following information was supplied regarding data availability:

The raw measurements are available in the Supplementary Files.

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
