# Peer review of "Evaluation of total immunoglobulin G and subclass antibodies in an enzyme-linked immunosorbent assay for serodiagnosis of human amebic liver abscess"

_PeerJ, doi:10.7717/peerj.14085_

## Round 0.1 · original submission · Major Revisions

The review process is now complete, and three reviews are included at the bottom of this letter (also see the annotated file). All reviewers and I agree that your manuscript deserves to be published. While the findings are original and relevant to the field, we identified some concerns and flaws that must be rigorously considered in your resubmission.
The experimental design and statistical analysis need to be carefully revised and modified accordingly. English language must be corrected and improved to make the manuscript clearer to the readers.

Reviewer 1 ·

Basic reporting

The authors submit an evaluation of total immunoglobulin G and subclass antibodies in an enzyme-linked immunosorbent assay for serodiagnosis of human amebic liver abscess. Evaluation of new diagnostic tests is essential. Thank you for taking this step.

The least important points;

2- Your introduction needs more detail. I suggest that you improve the description at lines 85- 93 to provide more justification for your study (specifically, you should elaborate on the advantages and disadvantages of using antibodies for the diagnosis of ALA). Could you please develop the problems of false positives in endemic areas. And shorten the first part (at lines 57-85).

3- At lines 75-76: The authors refer to the detection of "motile trophozoites" in ALA. However, molecular biology in liver abscesses is the gold standard (Zaman et al. 2000, Ryan et al. 2017) and the probability of finding motile forms on direct examination of pus is very low.

4- At lines 298-299: The authors recommend the use of the ELISA studied in this study as a routine test. The test appears to be time consuming to set up and use. How do the authors intend to use it routinely for the diagnosis of ALA? And compared to commercially available assays?

5- The use of names of parasitic pathologies should be improved. Some examples where the language could be improved include lines 124, 125, 128, 135, 222, 223, 224, 260, 261, 263, 265 – the current phrasing makes comprehension difficult. I suggest you to choose between the name of the disease or the use of the genus and species of the parasite.

6- Table 1. should be considered as supplemental data.

Experimental design

The most important issue;

1- Zaman et al. 2000 (doi : 10.1007/pl00008558), Ryan et al. 2017 (doi :10.1016/j.pt.2017.03.005), Ghelfenstein-Ferreira et al 2020 (doi :10.1128/JCM.01153-20) have shown that the ALA classification you use in Lines 119-122 is not the most appropriate for this situation, especially in endemic area. Please explain why you used this method rather than using positive PCR in the pus aspirate. The calculation of the analytical performance depends on this classification. Thus, it might be interesting to compare the Kappa correlation with antibody kit tests routinely use and the ELISA IgG1 assay.

The use of this new ELISA IgG-1 test seems interesting but referring to point 1 we cannot conclude on the performance of the test in the diagnosis of ALA.

Validity of the findings

no comment

Reviewer 2 ·

Basic reporting

.

Experimental design

.

Validity of the findings

.

Additional comments

This work is interesting because amebiasis considered a neglected disease. A
strength of this work is the amount of data acquired over many years of
collaboration between basic and clinical research groups.

As is, however, the paper would benefit from some major revisions, particularly
when it comes to the sample sizes and the focus of Group 3.

I suggest improving the manuscript with the following points

1.- One drawback is the disparity in the number of sera included in each group.

Group 2 includes only 14 sera. This is the most important group. The title of the
manuscript is mainly addressed to this group, so I suggest increasing the
number of ALA patient’s sera.

Table 1 shows demographic and clinical information of 14 amebic liver abscess
(ALA) patients. However, table 1 shows the inclusion of 4 patients without
clinical data confirming amoebic liver abscess. As this is the most important
group, each patient must be correctly identified. Only sera from patients with
evidence of amoebic abscess should be included.

Group 3 includes 254 samples, but this is not the most important group.
I suggest giving another approach to group 3 data. The study was carried out
using a bank of sera, so the authors do not rule out that the positives of Group 3
previously had invasive amebiasis. The authors are probably finding antibodies
from previous amebiasis, but the authors do not show adequate experiments to
demonstrate cross-reactions.

The authors should focus on the ALA group and perform the statistical analysis
mainly on group 2, and the paper would be strengthened if they increase the
sample.

It is worth wondering whether the test the authors developed can be used for
invasive amebiasis diagnosis among multi-parasite population such as Group 3.
This would be a useful application and perhaps the authors would like to
explore this in a separate paper focusing on this group and remove those
findings from the present paper.

2.- A major concern is the handling of the antigen used in ELISA tests lines 154
– 164.

The trophozoites were washed, then sonicated and centrifuged without using
proteinase inhibitors. Inhibitors were used starting from supernatant dialysis

A great obstacle in the development of diagnostic tools for amebiasis is the
elevated enzymatic activity of the amebic extracts. To prevent proteolytic
activity, enzymatic inhibitors are used. However, these inhibitors are not
completely effective and protein degradation continues. If these antigens are
used in ELISA tests, the absorbance index variation is high, as shown by Flores
et al., (Parasitology 2005, Vol 131, pp 231–236, and Experimental Parasitology
2016. Vol 161 pp 48-53).

The authors don’t show the intra assay coefficient obtained by measuring the
same sera in ten wells in the same plate. And the raw data doesn’t show it, nor
the absorbances of the duplicates.

I would recommend acceptance with major revision, as suggested above.

·

Basic reporting

The authors aimed to evaluate the diagnostic performance of IgG subclasses in the serodiagnosis of amoebic liver abscess (ALA). They used 308 sera, including 14 from Japanese patients with ALA, 40 from healthy Thai individuals, and 254 from Thai patients with other conditions. They then compared the sensitivity and specificity of ELISA targeting global IgG, IgG1, IgG2, IgG3 and IgG4, respectively. The introduction intelligibly describes the background, the knowledge gap, and the research question. The English language used is clear and unambiguous. The raw data are provided in a clear and well-structured manner.

- In Table 3, the bottom section labeled "IgG total and IgG subclass agreement test" needs to be clarified. I assume that the row "Agreement" is the proportion of observed agreement used to calculate Cohen's Kappa, that "Kappa" refers to Cohen's Kappa, and finally that "Prob>Z" is equivalent to pvalue.

Experimental design

The experimental design of the present study is more appropriate for assessing specificity than sensitivity since a large number of sera with other parasitic or liver infections are included to assess cross-reactions. Sensitivity is less well evaluated because of the small number of ALA patients included (n=14) as indicated in the discussion (line 286-287).

- In the instructions for the Bordier ELISA serological test for Entamoeba histolytica, it states, "Cross-reactions occur primarily in patients with leishmaniasis, malaria, filariasis, and strongyloidiasis." In your study, you did not include patients infected with leishmaniasis, malaria, and filariasis. It might be relevant to include sera from such patients. In addition, the main differential diagnosis of amoebic liver abscess (ALA) is pyogenic liver abscess. It would be beneficial to your study to include sera from such patients. If it is not possible for you to include such patients you should at least add a sentence in the limitation section of your discussion.

- For clarity, it could be possible to normalize the optical density (OD) of each sample by the corresponding cut-off value in Table 2 and Figure 1. An index equal to the sample OD divided by the cut-off OD should provide more relevant information than the raw OD (positive serology if index >1 et negative if index <1). It is just a suggestion.

- Characteristics of false-positive patients should be described, at least for total IgG and IgG1, including history of intestinal amoebosis, parasitological examinations of stool samples positive for Entamoeba histolytica/dispar if available.

- Calculations of positive predictive value and negative predictive value in Table 3 are irrelevant since reference patients are not selected to mirror the prevalence of the target population. Indeed, these two parameters are fonction of the prevalence of ALA in the target population.

Validity of the findings

The findings of this work are well formulated, limited to the original study question, and may be useful to other investigators in developing a new serological test for Entamoeba histolytica.

Additional comments

I commend the authors for their interesting and meaningful work that gives clues to develop other serological tests for ALA. This study could be strengthened with inclusion of sera from patients suffering from leishmaniasis, malaria, filariasis to broaden its external validity.

---

## Round 0.2 · Minor Revisions

Although most of the changes suggested by the reviewers have been made, the manuscript still needs revision. I agree with Reviewer#1 that would be great if the detailed responses in the Rebuttal Letter will be included in the manuscript.

Reviewer 1 ·

Basic reporting

no comment

Experimental design

no comment

Validity of the findings

no comment

Additional comments

Many thanks to the authors for taking the time to answer each point step by step. Thanks also to them for developing each of their answers with references and explanations. I would have liked the scientific community to also benefit from their response to our questions, specifically for point 1 and point 4.

Dear authors, could you please include your response in the appropriate sections (methods or discussion).
Thank you for your work.

·

Basic reporting

The authors answered my questions correctly, the limitations of the study are clearly stated in the discussion, and I think this manuscript is suitable for publication in PeerJ.

Experimental design

.

Validity of the findings

.

Additional comments

.

---

## Round 0.3 · accepted · Accept

The authors added all the information requested to the topics of the manuscript. Please, during the checking proofs, change "that cases" to "those cases" (line 131).